# Health, Occupational Stress, and Psychosocial Risk Factors in Night Shift Psychiatric Nurses: The Influence of an Unscheduled Night-Time Nap

**DOI:** 10.3390/ijerph20010158

**Published:** 2022-12-22

**Authors:** Valérie Amiard, Frédéric Telliez, Florine Pamart, Jean-Pierre Libert

**Affiliations:** 1Service de Santé au Travail, Centre Hospitalier Phillie Pinel, 80480 Dury, France; 2Institut d’Ingénierie de la Santé-UFR de Médecine, Université de Picardie Jules Verne, rue des Louvels, 80036 Amiens, France; 3Laboratoire PERITOX (UMR-I 01, Unité mixte INERIS), Centre Universitaire de Recherche en Santé, Présidence, Chemin du Thil, 80000 Amiens, France

**Keywords:** nurses, night work, night-time nap, psychosocial risk factors, health disorders

## Abstract

Background: Occupational stress and shift work (including night shift work) are associated with physical and psychological health consequences in healthcare providers in general and those working in psychiatric establishments in particular. The aim of this study was to assess the impact of occupational risk factors and unscheduled night-time naps on self-reported health disorders among nurses working in a French psychiatric hospital. Methods: We performed a 12-month observational field study of experienced nurses working at Philippe Pinel Psychiatric Hospital (Amiens, France) between September 2018 and September 2019. A comparative descriptive study of two groups of nurses who filled out a questionnaire on health and occupational stress was performed: nurses working permanently on the night shift (the night shift group, who took unscheduled naps), and nurses rotating weekly between morning and afternoon shifts (the day shift group). Results: The night and day shift groups comprised 53 and 30 nurses, respectively. There were no intergroup differences in health disorders, sleep quality, occupational stress, and risk factor perception. Correlation analyses showed that in the day shift group, a low level of support from supervisors was associated with elevated levels of distress, anxiety, and gastrointestinal disorders. In the night shift group, a greater overall work load was associated with elevated levels of anxiety and distress. These findings indicated that the nurses on the night shift had adapted well to their working conditions. Conclusions: An organizational strategy including an unscheduled night-time nap might improve health among night shift nurses.

## 1. Introduction

Psychological distress (i.e., anxiety and depression) is a common complaint among nurses in Western societies [1]. Demanding work tasks, poor social support and a low degree of job control (usually defined as job strain) are work stressors with a negative impact on health [2] and psychological well-being [3].

Occupational stress is known to be a major problem for healthcare workers in general [4] and for workers in psychiatric establishments in particular [5]. In psychiatric hospitals, nurses have to deal with emotional and psychological stress and a heavy physical work load [6]. It has been reported that (i) work stress factors in psychiatric departments are positively correlated with the nurses’ levels of depression, and (ii) the level of stress experienced at work is a significant predictor of depression among psychiatric nurses [7].

In psychiatric hospitals, nurses usually work either permanently on the night shift or rotate weekly between morning and afternoon shifts. It is known that in the absence of circadian adjustment, shift work (including night shift work) is associated with sleep disorders [8], an elevated frequency of cardiovascular and digestive diseases, and a greater risk of primary breast cancer [9,10]. Permanent night nurses are particularly prone to psychological stresses and burnout [11,12]. Night work also worsens nurses’ alertness and occupational performance, and thus decreases the quality of care [13].

The present study of nurses in a French psychiatric hospital sought to establish the impact of work stressors and psychological risks on the health of two groups of nurses: those working permanently on the night shift (the night shift group, who often took unscheduled naps) and those rotating weekly between morning and afternoon shifts (the day shift group, who rarely took unscheduled naps). To reduce psychological distress, a psychiatric nurse’s coping strategies are usually focused on cognitive factors. Kamal et al. (2012) [14], Nayomi, (2016) [15], Zaki (2016) [16], Hasan and Tumah (2018) [17], Tahghighi et al. (2017) [18] reported that further research should directly compare psychological outcomes and resilience in nurse shift workers vs. non-shift workers. Shift patterns can affect nurses’ sleep patterns and quality and thus pose risks to their health and safety. Particularly when it involves night shifts, shift work can result in sleep restriction, circadian desynchronization and fatigue, which in turn can have detrimental impacts on patient safety (due to errors and adverse events) and the quality of care [19].

In the present study, we hypothesized that taking an unscheduled night-time nap is a good coping strategy (based on physiological needs) for managing sleep debt and combating the harmful effects of night work on health outcomes in general and on psychological impacts in particular.

## 2. Materials and Methods

### 2.1. Design

We performed a 12-month, cross-sectional, observational study of the impact of night shift work and an unscheduled night-time nap on self-reported health disorders in experienced psychiatric nurses working at Philippe Pinel Psychiatric Hospital (Amiens, France) between September 2018 and September 2019. Questionnaire data were collected between February and May 2019.

### 2.2. Study Population and Setting

All the participants were volunteers and had undergone a medical evaluation of their physical and psychological aptitude to work in a psychiatric hospital. In the present study, we included nurses who had undergone the examination in the previous 12 months. Particular attention was paid to anxiety, depression, vigilance and sleep disorders. All subjects were in good general health and have not taken any medication which could disturb sleep pattern.

The participants in the night shift group worked from 21:15 to 6:30 for four consecutive nights in a row and then had two nights off. The day shift group rotated weekly between a morning shift (6:15 to 14:30) and an afternoon shift (14:15 to 21:30). For the same number of patients, the day shift comprised five nurses and the night shift comprised two nurses. During the night shift, each nurse typically slept for three to four hours (in a single episode) in a semi-reclining arm chair in a quiet, dark, thermally comfortable room. The time of this sleep episode was not planned in advance but depended on the night’s workload and/or agreement with the co-worker. This procedure was not official; in France, night-time naps at the workplace have not yet been widely introduced. The night nurses’ duties consisted in responding to the patients’ requests, administering prescribed drugs, changing incontinence products, and applying restraints when necessary. Under these circumstances, taking a very long nap is possible but the nurse may be woken up at any time for an emergency or to administer treatment. The hospital’s nurses may be confronted with violent and/or unexpected behaviour by the patients, although the emergency services deal with the most complex situations.

### 2.3. Ethical Considerations

The hospital’s administrative council, management staff, and health and safety committee approved the study protocol. In line with French legislation, approval by an independent ethics committee was not required. This study was nevertheless performed in accordance with the ethical standards of the 1964 Declaration of Helsinki and its subsequent revisions. All data were stored securely, in line with European Union’s General Data Protection Regulation and the guidelines issued by the French National Data Protection Commission (Commission nationale de l’informatique et des libertés (Paris, France)). The study questionnaire (see below) was also submitted to the hospital’s administrative council, management, and health and safety committee before the start of the study. Before filling out the questionnaire, all the participants provided their written consent after being comprehensively informed about the study’s objectives and procedures. By providing their consent, participants confirmed that they understood (i) the study information, (ii) that data collected for research purposes would remain confidential, and (iii) that they could contact the research team if they had any further questions. In return for their participation, participants received an oral presentation of the study results.

### 2.4. Instruments

We recorded the following sociodemographic characteristics: age, sex, the number of children living at home, marital status, the type of dwelling, the home-to-work travel time, and the number of years of shift work.

Sleep disturbances were self-reported according to criteria selected from a standard sleep questionnaire (total sleep duration, difficulties in falling asleep, sleep latency, and the number of spontaneous awakenings after falling asleep) and a set of questions on sleep quality and the restorative function of sleep (deep vs. light, calm vs. agitated, restful vs. ineffective, and difficulty waking up) from the Pittsburgh Sleep Quality Index [20]. Each parameter was scored on a visual analogue scale bounded by each pair of adjectives and divided into four quarters labelled as “very good”, “fairly good”, “fairly bad” and “bad”. Values of 0 (the best possible situation), 33, 66 and 100 (the worst possible situation) were respectively attributed to the four adjectives.

Work characteristics were assessed with the French National Health and Safety Institute questionnaire [21], which is based on that developed by Haims and Carayon (1988) [22] and has been psychometrically validated by Carayon et al. (1998) [23]. As recommended by the developers, the questionnaire was completed in the presence of the investigator. This questionnaire quantifies the respondent’s occupational psychological risk in general and their physical symptoms of stress in particular. There are three items on the work load in general, three on the current work load, four on work pressure, two on attention required to perform tasks, five on job control, three on decision-making relative to the person’s role in the work organization, four on support from supervisors and colleagues, and two on confidence in the person’s professional future. Each factor is measured on a 4- or 5-point scale. The scores were calculated on a scale graduated from 0 to 100 and averaged according to the number of questions. On a 5-point scale, for example, item 1 was coded as 0, with item 2 coded as 25, item 3 coded as 50, item 4 coded as 75, and item 5 coded as 100; the higher the calculated score, the higher the level of occupational psychosocial risk. Lastly, in order to test the night nurses’ level of motivation, all had to answer the following question: “Would you accept an offer to work the day shift?”.

The questionnaire also probed self-reported health disorders (cardiac risk factors, anxiety, distress, and digestive disorders) over the previous 12 months. There were two questions on cardiac risk factors (palpitations and precordial pain), three questions on distress (the presence or absence of sweating at rest, nervousness or trembling, and fainting), seven on anxiety (anxiety, irritability, insomnia, intense fatigue or exhaustion, tenseness, depression, and difficulties falling asleep), and five on digestive disorders (bloating, flatulence, poor digestion, constipation or diarrhea, a knotted feeling in the stomach, and mouth dryness). For each item, the answers were scored as 1 (never or rarely), 2 (sometimes), 3 (often enough) and 4 (very often or constantly) by each participant and then converted to a score of 0, 33, 66, and 100, as described above. The item scores were summed; the higher the overall score, the poorer the respondent’s physical health. All the questionnaires were anonymized prior to analysis.

### 2.5. Statistical Analysis

Variables were reported as the mean ± standard deviation (SD) or the median [range], as appropriate. Statistical analyses were performed with Statistica software (version 7.1, TIBCO Software Inc., Palo Alto, CA, USA). Intergroup comparisons were performed with non-parametric, unpaired Mann–Whitney U tests. Spearman’s correlation coefficient (*r*) was used to assess the correlation between self-reported health disorders and occupational stress-related risk factors. Spearman’s *r* was 0–0.3 for a weak correlation, 0.3–0.6 for a moderate correlation, and >0.6 for a strong correlation. The threshold for statistical significance was set to *p* ≤ 0.05.

## 3. Results

A total of 83 hospital psychiatric nurses participated in the survey: 30 in the day shift group and 53 in the night shift group. Most of the respondents were females (76% and 74% in the day and night shift groups, respectively). The mean ± SD age was similar in the two groups (38.9 ± 17.7 in the day shift group vs. 36.8 ± 11.3 in the night shift group), as was the number of years of shift work (12.0 ± 21.7 vs. 14.6 ± 22.2, respectively). There was no intergroup difference in the body mass index (23.1 ± 3.0 in the day shift group vs. 23.3 ± 2.9 in the night shift group), the proportion of smokers (50% vs. 40%, respectively) or the duration of tobacco use (6.0 ± 7.7 vs. 5.6 ± 7.5 years, respectively).

The mean home-to-work travel time was similar in the two groups (22.7 ± 12.5 min in the day shift group vs. 24.1 ± 18.7 in the night shift group). With regard to the level of motivation, eight nurses (27%) in the day shift group and none in the night shift group wanted to change their shift.

The two groups did not differ significantly with regard to family-related factors: 77% of the day shift nurses and 90% of the night nurses were married, while the proportions with working spouses were, respectively, 90% and 92%. Likewise, the day shift and night shift groups did not differ in terms of the mean number of children over the age of 10 (0.1 ± 0.4 vs. 0.3 ± 0.6, respectively) or under the age of 10 (0.7 ± 0.8 vs. 1.0 ± 0.7, respectively).

The self-reported sleep quality scores did not differ significantly when comparing the two groups (Table 1). With regard to self-reported health disorders, there were no significant differences between the day shift and night shift groups in the scores for cardiovascular disorders (11.5 ± 14.0% vs. 16.5 ± 17%, respectively; *p* = 0.167), gastrointestinal disorders (22.9 ± 16.0% vs. 17.6 ± 19.0%; *p* = 0.193), anxiety (31.3 ± 16.0% vs. 26.0 ± 17%; *p* = 0.168) or distress (16.5 ± 11.0% vs. 14.6 ± 13.0%; *p* = 0.750).

There were no intergroup differences in most of the occupational stress risk factors (Table 2). The scores for perceived support from supervisors and from colleagues were slightly higher (i.e., corresponding to greater constraints) in the day shift group, although the differences were not statistically significant (*p* = 0.052 and 0.056 for support from supervisors and from colleagues, respectively).

A correlation analysis showed that support from supervisors had a significant effect in the day shift group only: the lower the level of support, the higher the levels of distress (*r* = −0.37, *p* = 0.047), anxiety (*r* = −0.42, *p* = 0.020) and gastrointestinal disorders (*r* = −0.47, *p* = 0.009).

In the night shift group only, a low perceived level of work control was associated with high levels of distress (*r* = −0.32, *p* = 0.020) and anxiety (*r* = −0.24, *p* = 0.067). In this latter group, the current work load was significantly correlated (*r* = 0.43, *p* = 0.001) with cardiac palpitations.

In both groups, all the other occupational stress factors (work pressure, attention required for work tasks, decision-making in the work organization, confidence in the future, and support from colleagues) were not significantly correlated with health disorders.

## 4. Discussion

Comparisons of the literature studies are complicated by differences in the types of shift work and in the methods used to measure psychological outcomes. In the present study, we used a standardized analysis and a direct comparison of permanent night shift and nurses working during daytime hours; this is one of the strengths of our study. Around half of the literature studies compared different types of shift working rather than shift workers vs. non-shift workers.

In the present study, night shift nurses and daytime shift nurses did not differ markedly overall with regard to self-evaluated psychological and health disorders. Nurses in the night shift group did not report persistent poor sleep quality when questioned about difficulties in initiating and maintaining sleep and difficulty awakening—the main criteria used to code sleep disorders related to shift work. Our results suggest that the night nurses had learned to cope with night work and did not suffer from marked health disorders as a result of night working. The self-reported sleep scores and total sleep time over a 24 h period were similar in the two groups, suggesting that the night nurses made up for the accumulated night-time sleep debt during the day. These similarities might also explain why we did not observe self-reported sleep disturbances and cardiovascular disorders in the night shift group, confirming the results reported by Petrov et al. (2014) [24].

The nurses’ night-time sleep strategy is the most plausible explanation for this limited impact. As reported by Silva-Costa et al. (2015) [25], nurses who take a long nap during the night shift require 40% less sleep during the following morning. Taking a nap during the night can reduce the adverse effects of night work by sustaining circadian rhythms, which also helps to compensate for sleep loss [26]. Similarly, Asaoka et al. (2013) [27] suggested that along with long night-time working hours and perturbed circadian rhythms, missing napping opportunities during night work is associated with the occurrence of shift work disorders.

The beneficial effect of napping has also been reported by several other researchers [28,29,30,31,32,33,34,35]. Furthermore, Burdelak et al. (2012) [36] determined that permanent night nurses who napped reported fewer complaints than non-permanent night nurses and nurses working on rotating day shifts. However, these conclusions are subject to debate because the beneficial effects of napping were not observed when sleep was measured objectively during a simulated night in the laboratory [37].

One disadvantage of a long period of sleep during the night shift is sleep inertia, which can decrease the operator’s vigilance levels after awakening. In the present study, sleep inertia was not relevant because none of the care activities were carried out alone or involved complex technical procedures that might lead to major errors in patient care.

The two groups differed slightly with regard to the correlations between health disorders and occupational psychosocial risk factors. Higher levels of anxiety and distress were correlated with a lack of support by supervisors in the day shift group and with low perceived control in the night shift group. This result can probably be explained by the characteristics of the work in the day vs. night periods. In this study, negative perception of the support by supervisors appeared to reflect stressful experiences in the workplace during the day shift only (with associated digestive disorders), while negative perception of the work load during night was associated with cardiac palpitations.

The present field study had several limitations. Firstly, we studied highly motivated volunteers who had chosen day shift or night shift work for family reasons; this might have caused the participants to understate the extent of their health disorders [28,38,39]. Secondly, the respondents were relatively young. Koller (1983) [40] and Walsh et al. (1987) [41] have reported that the prevalence of health disorders increases with age and is especially high in the 40 to 50 age group. According to Cervinka (1993) [42], permanent night nurses tend to experience severe health disorders after more than 20 years on shift. The nurses in the present study were highly experienced and had been working shifts for many years (an average of 12.0 and 14.6 years in the day shift and night shift groups, respectively) and probably tolerated this work schedule well, thus weakening the stress–health disorder relationship [43,44]. Self-selection effects are frequently encountered in health research on shift [45] because nurses who do not tolerate shift work tend to abandon this work system. The present study also had a number of strengths. The high proportion of females in the study population (74%) was representative of psychiatric nurses in France. The two study groups were also similar with regard to their educational background, family factors, sex distribution, age distribution, and work seniority.

The results of the present real-life study showed that (i) shift work is not associated with worse psychological functioning or lower resilience in nurses (this confirms the integrative review by Tahghighi et al. (2019) [46]). However, this unscheduled organizational strategy is not widespread in psychiatric hospitals. The supervisors’ lack of knowledge about and attitudes to napping are major obstacles [47].

This unscheduled work organization can be only implemented when the workload is discontinuous and characterized by long rest periods. Hence, unscheduled night-time napping should be considered as an opportunity and a factor that improves human performance, health, safety, and quality of care. It is doubtless important to define organizational barriers and facilitators with nurse managers.

Our results might also have significant implications for shift workers in other industrial and health sectors. In fact, as reported by Capadona et al. [19], the literature data suggest that workers who find night shift work particularly hard have higher accident rates. Therefore, workplace initiatives of such night-time napping might raise awareness of the importance of sleep and minimize the impact of fatigue in the workplace. However, the introduction of napping in the workplace might also have adverse effects when a rapid, accurate response is required (i.e., in critical care unit). It is well known that hypnic inertia after nocturnal awakening induces a hypovigilance state lasting from 10 to 20 min; this might affect the worker’s reaction time and impair their decision-making ability.

To the best of our knowledge, there are no guidelines on unscheduled night-shift napping by nurses in France or other developed countries. This practice is informally (but not formally) accepted by hospital authorities. An informal arrangement is possible with the agreement and collaboration of nurse leaders and will depend on the type of activity. We now know enough about night-time napping to enable nurse leaders to implement and organize the process and to help nurses protect their health and the quality and safety of care. 

## 5. Conclusions

The results of the present study showed that (i) shift work is not associated with self-reported health disorders and (ii) permanent night nurses had adapted well to their work schedule. Unscheduled night-time napping might represent an active coping strategy in which the nap’s time and duration are chosen as a function of physiological requirements. In hospital departments, decision-makers should facilitate this sleep/wake scheduling by notably providing staff with access to comfortable napping rooms.

## Figures and Tables

**Table 1 ijerph-20-00158-t001:** Median (range) scores for sleep disorders, as reported by the day shift and night shift groups. None of the intergroup differences were statistically significant. The scores were assessed on a scale graduated in four quarters from 0 (very good) to 100 (very bad). Please refer to the text for details. The higher the score, the lower the quality of the sleep or the waking-up status.

	Day Shift Group (*n* = 30)	Night Shift Group (*n* = 53)
Total sleep time (hours)	7 (5–10)	7 (4–12)
Sleep latency (min)	15 (0–60)	30 (0–75)
Difficulty falling asleep	33 (0–33)	33 (0–66)
Number of awakenings per night (n)	2 (0–9)	2 (0–5)
Subjective sleep quality	33 (0–100)	33 (0–100)
Difficulty waking up	66 (0–100)	66 (0–100)
Good physical shape on awakening	66 (0–100)	66 (0–100)
Difficulty waking up during rest periods	0 (0–100)	0 (0–66)

**Table 2 ijerph-20-00158-t002:** Median (range) self-reported scores for occupational stress risk factors (from the French National Research and Safety Institute questionnaire) in the day shift and night shift groups. The scores were assessed on a 0-to-100 scale: the higher the score, the higher the level of the occupational psychosocial risk factor.

	Day Shift Group (*n* = 30)	Night Shift Group (*n* = 53)
Work load in general	44.0 (11–88.7)	66.0 (33–100)
Current work load	33.3 (0–100)	33.3 (0–100)
Work pressure	24.8 (0–66)	33.0 (0–66.5)
Attention	66.5 (0–100)	83.0 (49.5–100)
Work control	48.2 (0–78.2)	56.6. (0–80)
Involvement in work organization	50.0 (16.7–83.3)	50.0 (0–91.7)
Support from supervisors	33.0 (0–91.5)	16.5 (0–91.5)
Support from colleagues	16.5 (0–41.3)	8.3 (0–41.3)
Career confidence	16.5 (0–66.5)	16.5 (0–66.5)

## Data Availability

Not applicable.

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
