# Peer review of "Health, Occupational Stress, and Psychosocial Risk Factors in Night Shift Psychiatric Nurses: The Influence of an Unscheduled Night-Time Nap"

_ijerph, 2022, doi:10.3390/ijerph20010158_

Round 1
Reviewer 1 Report
This is a well-written original look at unscheduled night shift napping for nurses. The discussion/conclusions might benefit from more detail about the norms of taking such long on-shift naps. This seems quite atypical - it certainly is not happening in U.S. hospitals. The final conclusions that "(i) shift work is not associated with worse psychological functioning or lower resilience in nurses (thus 245 confirms the integrative review by Tahghighi et al. (2019) [45]), and (ii) permanent night nurses had adapted well to their work schedule" should be presented in light of these highly unusual sleep opportunities which influence the ability to generalize findings. The authors might discuss the need for other organizations to consider expanding the napping ability of their night shift workers. This was a very interesting read.
Author Response
Responses to Reviewers ijerph-2068172
Health, Occupational Stress, and Psychosocial Risk Factors in Night Shift Psychiatric Nurses: the Influence of an Unscheduled Night-time Nap.
Valérie Amiard1, Frédéric Telliez2,3*, Florine Pamart2 and Jean-Pierre Libert3
Response to reviewer
December 9h, 2022
and
To the Editor IJERPH,
Dear Guest Editors of special Issue
« Health consequences of shift work
and chronobiological disruption »,
Dear Reviewer,
Thanks a lot for your reviewing and for your comments and positive criticisms.
We have paid attention to your comments and advice for improving the manuscript.
See below our responses to each point.
Please find our revision version of our manuscript by Amiard et al. entitled ““Health, occupational stress, and psychosocial risk factors in night shift psychiatric nurses: the influence of an unscheduled night-time nap”, for consideration.
We hope that our work is worthy of publication in your Special Issue of IEJHRP.
Sincerely yours,
Frédéric Telliez, Jean-Pierre Libert, Valérie Amiard, on behalf of the co-authors.
Corresponding author: Frédéric Telliez, frederic.telliez@u-picardie.fr
Reviewer 1
This is a well-written original look at unscheduled night shift napping for nurses. The discussion/conclusions might benefit from more detail about the norms of taking such long on-shift naps. This seems quite atypical - it certainly is not happening in U.S. hospitals.
We now discuss this point in the last paragraph of Discussion section, at line
To the best of our knowledge, there are no guidelines on unscheduled night-shift napping by nurses in France or other developed countries. This practice is informally (but not formally) accepted by hospital authorities. An informal arrangement is possible with the agreement and collaboration of nurse leaders and will depend on the type of activity. We now know enough about night-time napping to enable nurse leaders to implement and organize the process and to help nurses protect their health and the quality and safety of care.
The final conclusions that "(i) shift work is not associated with worse psychological functioning or lower resilience in nurses (thus 245 confirms the integrative review by Tahghighi et al. (2019) [45]), and (ii) permanent night nurses had adapted well to their work schedule" should be presented in light of these highly unusual sleep opportunities which influence the ability to generalize findings. The authors might discuss the need for other organizations to consider expanding the napping ability of their night shift workers. This was a very interesting read.
We now discuss this comment in the Discussion:Line 278-291
This unscheduled work organization can be only implemented when the workload is discontinuous and characterized by long rest periods. Hence, unscheduled night-time napping should be considered as an opportunity and a factor of that improves human performance, health, safety, and quality of care. It is doubtless important to define organizational barriers and facilitators with nurse managers.
Our results might also have significant implications for shift workers in other industrial and health sectors. In fact, as reported by Capadona et al. [19], the literature data suggest that workers who find night shift work particularly hard have higher accident rates. Therefore, workplace initiatives such night-time napping might raise awareness of the importance of sleep and minimize the impact of fatigue in the workplace. However, the introduction of napping in the workplace might have also adverse effects when a rapid, accurate response is required (i.e.; in critical care unit). It is well known that hypnic inertia after nocturnal awakening induces a hypovigilance state lasting 10 to 20 minutes; this might affect the worker’s reaction time and impair his/her decision-making ability.

Reviewer 2 Report
General comments
Interesting article on health consequences of shift work in nurses.
Some changes are suggested, mainly related to the redaction of the introduction, material and methods and conclusions.
Specific comments
Abstract: The date when the work was carried out should be given in the abstract and main text. Inclusion and exclusion criteria must be specified.
Introduction: A deeper approach to the relationship between shift work in nursing, the desynchronization of the circadian rhythm and patient safety would be recommended. The introduction is too poor in relation to previous works that analyze this topic.
Matherial and Methods: Inclusion and exclusion criteria must be specified. It is necessary to specify the dates of the study.
The information is not correctly structured. The following sections should be specified:
· Study population and Setting.
· Design.
· Instruments.
Lines 93-94: The bibliographical reference is presented in two formats “[19] (Buysse et al., 1989)”.
Line 99: The bibliographical reference is presented in two formats “[20] (Cail et al., 2000)”.
Results:
Lines 172-177: Inverse correlations are shown. If so, they do not correlate with the correlation coefficients shown (sign).
Discussion: The discussion section is well done.
Conclusions: Avoiding discussion with other papers (line 246) in this section, which should be in the previous section. The authors cite “ The supervisors’ lack of knowledge about and attitudes to napping are major obstacles (Geiger-Brown et al., 2016) [46]”. Is this conclusion your own or from [46]? Has this item been studied? If so, this study variable does not appear. Nevertheless, the bibliographical reference is presented in two formats.
Institutional Review Board Statement: Even if French law does not provide for approval by an independent ethics committee, it is necessary to clarify the ethical principles governing the study and European legislation on data protection.
References: The bibliography is complete, current and well formatted.
Author Response
Responses to Reviewers ijerph-2068172
Health, Occupational Stress, and Psychosocial Risk Factors in Night Shift Psychiatric Nurses: the Influence of an Unscheduled Night-time Nap.
Valérie Amiard1, Frédéric Telliez2,3*, Florine Pamart2 and Jean-Pierre Libert3
Response to reviewer
December 9h, 2022
and
To the Editor IJERPH,
Dear Guest Editors of special Issue
« Health consequences of shift work
and chronobiological disruption »,
Dear Reviewer,
Thanks a lot for your reviewing and for your comments and positive criticisms.
We have paid attention to your comments and advice for improving the manuscript.
See below our responses to each point.
Please find our revision version of our manuscript by Amiard et al. entitled ““Health, occupational stress, and psychosocial risk factors in night shift psychiatric nurses: the influence of an unscheduled night-time nap”, for consideration.
We hope that our work is worthy of publication in your Special Issue of IEJHRP.
Sincerely yours,
Frédéric Telliez, Jean-Pierre Libert, Valérie Amiard, on behalf of the co-authors.
Corresponding author: Frédéric Telliez, frederic.telliez@u-picardie.fr
Reviewer 2
General comments
Interesting article on health consequences of shift work in nurses.
Some changes are suggested, mainly related to the redaction of the introduction, material and methods and conclusions.
Specific comments
Abstract: The date when the work was carried out should be given in the abstract and main text. Inclusion and exclusion criteria must be specified.
We now give more details in the Materials and Methods and the date was reported in the abstract Line 18
We performed a 12-month observational field study of experienced nurses working at Philippe Pinel Psychiatric Hospital (Amiens, France) between September 2018 and September 2019.
Introduction: A deeper approach to the relationship between shift work in nursing, the desynchronization of the circadian rhythm and patient safety would be recommended. The introduction is too poor in relation to previous works that analyze this topic.
We have added the following paragraph at line 60. We also discuss the importance of work performance and errors in the Discussion and Conclusions.
Shift patterns can affect nurses’ sleep patterns and quality and thus pose risks to their health and safety. Particularly when it involves night shifts, shift work can result in sleep restriction, circadian desynchronization, and fatigue, which in turn can have detrimental impacts on patient safety (due to errors and adverse events) and the quality of care [19].
Matherial and Methods: Inclusion and exclusion criteria must be specified. It is necessary to specify the dates of the study.
The information is not correctly structured. The following sections should be specified:
We have taken into account the reviewer’s advice and now provide these details in the Material and Methods section.
Design
We performed a 12-month, cross-sectional, observational study of the impact of night-shift work and an unscheduled night-time nap on self-reported health disorders in experienced psychiatric nurses working at Philippe Pinel Psychiatric Hospital (Amiens, France) between September 2018 and September 2019. Questionnaire data were collected between February and May 2019.
Study population and setting
All the participants were volunteers and had undergone a medical evaluation of their physical and psychological aptitude to work in a psychiatric hospital. In the present study, we included nurses who had undergone the examination in the previous 12 months. Particular attention was paid to anxiety, depression, vigilance and sleep disorders. All subjects were in good general health and were not taken any medication which can disturb sleep pattern.
The participants in the night shift group worked from 21:15 to 6:30 for four consecutive nights in a row and then had two nights off. The day shift group rotated weekly between a morning shift (6:15 to 14:30) and an afternoon shift (14:15 to 21:30). For the same number of patients, the day shift comprised five nurses and the night shift comprised two nurses. During the night shift, each nurse typically slept for three to four hours (in a single episode) in a semi-reclining arm chair in a quiet, dark, thermally comfortable room. The time of this sleep episode was not planned in advance but depended on the night’s workload and/or agreement with the co-worker. This procedure was not official; in France, night-time naps at the workplace have not yet been widely introduced. The night nurses’ duties consisted in responding to the patients’ requests, administering prescribed drugs, changing incontinence products, and applying restraints when necessary. Under these circumstances, taking a very long nap is possible but the nurse may be woken up at any time for an emergency or to administer treatment. The hospital’s nurses may be confronted with violent and/or unexpected behaviour by the patients, although the emergency services deal with the most complex situations.
Ethical considerations
The hospital’s administrative council, management staff, and health & safety committee approved the study protocol. In line with French legislation, approval by an independent ethics committee was not required. This study was nevertheless performed in accordance with the ethical standards of the 1964 Declaration of Helsinki and its subsequent revisions. All data was stored securely, in line with European Union’s General Data Protection Regulation and the guidelines issued by the French National Data Protection Commission (Commission nationale de l'informatique et des libertés (Paris, France)). The study questionnaire (see below) was also submitted to the hospital’s administrative council, management, and health & safety committee before the start of the study. Before filling out the questionnaire, all the participants provided their written consent after being comprehensively informed about the study’s objectives and procedures. By giving their consent, participants confirmed that they understood (i) the study information, (ii) that data collected for research purposes would remain confidential, and (iii) that they could contact the research team if they had any further questions. In return for their participation, participants received an oral presentation of the study results.
- Instruments
Statistical analysis
Lines 93-94: The bibliographical reference is presented in two formats “[19] (Buysse et al., 1989)”.
Line 99: The bibliographical reference is presented in two formats “[20] (Cail et al., 2000)”.
Thanks. We have corrected the citation.
Results:
Lines 172-177: Inverse correlations are shown. If so, they do not correlate with the correlation coefficients shown (sign).
Thank you for pointing out these mistakes, which we have corrected. Negative signs have been added to the coefficients for inverse correlations.
Discussion: The discussion section is well done.
Conclusions: Avoiding discussion with other papers (line 246) in this section, which should be in the previous section. The authors cite “ The supervisors’ lack of knowledge about and attitudes to napping are major obstacles (Geiger-Brown et al., 2016) [46]”. Is this conclusion your own or from [46]? Has this item been studied? If so, this study variable does not appear. Nevertheless, the bibliographical reference is presented in two formats.
In response to this comment, we have avoided discussion in the Conclusions section.
The supervisors’ lack of knowledge about and attitudes to napping are major obstacles [46].
This is the conclusion of Geiger-brown et al. 2016). This point has not been studied in the present study
Institutional Review Board Statement: Even if French law does not provide for approval by an independent ethics committee, it is necessary to clarify the ethical principles governing the study and European legislation on data protection.
We agree with these comments and we have clarified and detailed this aspect in Material and Methods:
The hospital’s administrative council, management staff, and health & safety committee approved the study protocol. In line with French legislation, approval by an independent ethics committee was not required. This study was nevertheless performed in accordance with the ethical standards of the 1964 Declaration of Helsinki and its subsequent revisions. All data was stored securely, in line with European Union’s General Data Protection Regulation and the guidelines issued by the French National Data Protection Commission (Commission nationale de l'informatique et des libertés (Paris, France)). The study questionnaire (see below) was also submitted to the hospital’s administrative council, management, and health & safety committee before the start of the study. Before filling out the questionnaire, all the participants provided their written consent after being comprehensively informed about the study’s objectives and procedures. By giving their consent, participants confirmed that they understood (i) the study information, (ii) that data collected for research purposes would remain confidential, and (iii) that they could contact the research team if they had any further questions. In return for their participation, participants received an oral presentation of the study results.
References: The bibliography is complete, current and well formatted.
